# Breast Cancer: Clinical–Epidemiological Profile and Toxicities of Women Receiving Treatment with Taxanes in the Amazon Region

**DOI:** 10.3390/jpm13101458

**Published:** 2023-09-30

**Authors:** Marta Solange Camarinha Ramos Costa, Marianne Rodrigues Fernandes, Esdras Edgar Batista Pereira, Diana Feio da Veiga Borges Leal, Rita de Cássia Calderaro Coelho, Elisa da Silva Menezes, Antônio André Conde Modesto, Paulo Pimentel de Assumpção, Rommel Mario Rodriguez Burbano, Sidney Emanuel Batista dos Santos, Ney Pereira Carneiro dos Santos

**Affiliations:** Núcleo de Pesquisa em Oncologia, Universidade Federal do Pará, Belém 66073-005, PA, Brazil; martasolange@ufpa.br (M.S.C.R.C.); edgarpereira@ufpa.br (E.E.B.P.); dianafeio@hotmail.com (D.F.d.V.B.L.); rccalderarocoelho@gmail.com (R.d.C.C.C.); elisamenezes19@gmail.com (E.d.S.M.); antonioacm@ufpa.br (A.A.C.M.); assumpcaopp@gmail.com (P.P.d.A.); rommelburbano@gmail.com (R.M.R.B.); sidneysantos@ufpa.br (S.E.B.d.S.); npcsantos.ufpa@gmail.com (N.P.C.d.S.)

**Keywords:** breast cancer, epidemiology, risk factors, toxicity, chemotherapy, taxanes

## Abstract

Breast cancer is the most common malignant disease and the leading cause of mortality among women worldwide. Antineoplastic chemotherapy is one of its primary treatments, typically based on the class of drugs known as taxanes. Despite their proven therapeutic efficacy, these drugs can induce severe toxicities, leading to dose limitations or even treatment discontinuation. The objective of this study was to describe the clinical–epidemiological profile, risk factors, and toxicities of taxane-based chemotherapy treatment in women with breast cancer in the Amazon region. This is a cross-sectional, quantitative, and descriptive study conducted with 300 women diagnosed with breast cancer undergoing taxane treatment. Most patients were in the 40–49 age range, of brown ethnicity, and had completed elementary school. The majority of patients had risk factors such as alcoholism and a sedentary lifestyles. Most women had their first pregnancy between the ages of 18 and 21, breastfed their children, had menarche between the ages of 12 and 13, and were pre-menopausal and with a family history of cancer. The most frequent histological type was non-special invasive carcinoma and the Luminal B subtype. Most participants in this study showed taxane toxicity, with neurotoxicity being the most frequent. These findings reveal the importance of early detection, comprehensive risk factors, and effective management of treatment toxicities to improve patient outcomes in breast cancer care in the Amazon region.

## 1. Introduction

Breast cancer (BC) is the most prevalent malignancy among women globally. Each year, more than two million new cases of breast cancer are diagnosed in women worldwide. In Brazil, it is estimated that there were 73,610 new cases of BC annually, accounting for 29.7% of all new cancer diagnoses in women. In the Brazilian Amazon region, breast cancer was the most frequently diagnosed cancer, with 2410 new cases per year during the triennium 2023–2025 [1].

Breast cancer risk is influenced by various factors, including age, genetics, family history, and radiation exposure. Around 90% to 95% of cases are sporadic, and not directly related to inherited genetic mutations. Instead, they are associated with factors like early menstruation, late menopause, nulliparity, late first childbirth (after 30), hormone therapy, postmenopausal obesity, physical inactivity, alcohol consumption, and smoking [2].

Regarding the histology of malignant breast tumors, the most commonly used classification today is from the World Health Organization and is based on the growth pattern and cytological characteristics of invasive tumor cells, which is divided into three specific groups: invasive carcinoma of no special type (Invasive Ductal Carcinoma—IDC), invasive carcinomas of special type, and rare types. Among these, invasive carcinoma of no special type is the most common, accounting for 40 to 75% of all invasive breast carcinomas [3]. For molecular classification, according to the St. Gallen Consensus Classification 2013, breast cancer can currently be divided into different subtypes, the main ones being Luminal A, Luminal B (HER2 positive or negative), HER2+, and triple-negative [4].

Paclitaxel and docetaxel are taxane antineoplastic agents commonly used as first-line treatments for breast cancer. These drugs exhibit significant efficacy in the treatment of breast cancer. However, they are associated with dose-limiting adverse events such as myelosuppression, peripheral sensory neuropathy, anaphylactic reactions, and the development of drug resistance [5]. Despite the use of a premedication regimen, neurotoxicity remains the primary dose-limiting toxicity of breast cancer therapy [6].

To classify toxicities induced by antineoplastic treatment, the National Cancer Insti-tute (NCI) and the National Institutes of Health (NIH) have developed the Common Ter-minology Criteria for Adverse Events (CTCAE). This consists of descriptive terminology that can be used for reporting adverse events related to oncology treatment in various pa-tient scenarios, especially in clinical research. According to CTCAE version 5.0, the grad-ing of antineoplastic toxicities, considering the severity of the event, can be defined as grade 1 for mild toxicity, asymptomatic, or mild symptoms; grade 2 for moderate; grade 3 for severe; grade 4 for life-threatening, and grade 5 referring to death related to the adverse event [7]. When a patient experiences any type of toxicity in grades 3 and 4 according to CTCAE, dose reduction or even treatment suspension is typically recommended.

Breast cancer is recognized as a global public health concern. Nevertheless, there are significant variations in breast cancer incidence across different regions worldwide. The risk factors and disease patterns associated with breast cancer patients in Europe may differ from those in South American countries. Therefore, it is crucial to urgently explore these geographic variations, risk factors, and disease characteristics that are specific to each region [8].

This study aimed to examine the clinical-epidemiological profile, risk factors, and treatment toxicity among women with breast cancer who underwent taxane therapy in the Brazilian Amazon region.

## 2. Materials and Methods

This descriptive, cross-sectional, and quantitative study was conducted on a sample of three hundred female breast cancer patients admitted to two public hospitals providing high-complexity oncology care in Belém, located in the Brazilian Amazon Region. The participants received antineoplastic chemotherapy treatment with taxanes and were selected between September 2020 and August 2021.

### 2.1. Ethical Aspects

The research protocol employed in this study received approval from the Committee for Research Ethics of the Federal University of Pará (approval no. 4209355). Prior to their participation in, each individual was thoroughly briefed about the research’s objectives. They were then asked to sign the study Informed Consent Form (ICF), which outlined the study’s procedures and assured the absolute confidentiality of their identity throughout the research process.

### 2.2. Chemotherapy Regimens

All the patients in this study received a combination of chemotherapy, first with Adriamycin and cyclophosphamide for 4 cycles, followed by taxanes in monotherapy with paclitaxel or docetaxel. Within the treatment profile, patients in the Paclitaxel group used the Adriamycin 60 mg/m^2^ and cyclophosphamide 600 mg/m^2^ protocol (every 21 days, for 4 cycles), followed by paclitaxel in monotherapy, at a dose of 80 mg/m^2^ (weekly, for 12 weeks) or 175 mg/m^2^ (every 21 days, for 4 cycles). As for the Docetaxel group, patients followed the protocol that included Adriamycin 60 mg/m^2^ and cyclophosphamide 600 mg/m^2^ (every 21 days, for 4 cycles), followed by docetaxel in monotherapy, at a dose of 35 mg/m^2^ (weekly, for 12 weeks) or 100 mg/m^2^ (every 21 days, for 4 cycles).

### 2.3. Data Collection

In this study, clinical profile data (histological type, molecular subtype, and disease progression) were extracted from patients’ records, while clinical-epidemiological profile data (age, ethnicity, marital status, education, occupation, lifestyle habits, reproductive profile, and family history) were obtained through interviews with the patients. con-ducted during the chemotherapy treatment period, shortly before or during the Taxane chemotherapy infusion.

Toxicity assessments of the patients were conducted at the chemotherapy service it-self, regularly at each 21-day cycle or every 4 weeks in the case of weekly Taxane protocols. This included a complete blood count assessment, which was routinely performed before the start of each cycle. During this time, patients were asked about any signs and symp-toms they had experienced during that period.

The emphasis of this study centered on monitoring the occurrence of signs and symptoms reported by patients as indicators of toxicity throughout exclusive Taxane chemotherapy. Data on interventions used to manage toxicities were not collected, as standardized clinical practices provided by the institutions were employed to control pa-tients’ side effects.

The interview questionnaire encompassed the collection of sociodemographic infor-mation and risk factors for a comprehensive characterization. We assessed the intensity of adverse events arising from anticancer treatments by utilizing the Common Terminology Criteria for Adverse Events (CTCAE—Version 5.0) [7].

### 2.4. Inclusion and Exclusion Criteria

All patients were females, aged 18 or older, had histologically confirmed with breast cancer, and were undergoing adjuvant or neo-adjuvant antineoplastic chemotherapy treatment with Taxanes. Male patients, those under 18 years of age, individuals with an ECOG Perfor-mance Status Scale of 4 or greater, those without a diagnosis of breast cancer, those with-out an indication for Taxane treatment, and those who did not agree to participate in the study or did not sign the ICF were excluded.

Toxicity signs or symptoms reported before or during the initial phase of treatment with the Adriamycin and Cyclophosphamide protocol were not included in this study. Adverse reactions that occurred before the initiation of taxane chemotherapy, such as al-opecia, were not considered in this study because they were also associated with other antineoplastic agents previously administered to the patients, such as Adriamycin and Cyclophosphamide. This was done to prevent erroneous conclusions regarding the exclu-sive toxicities of taxanes.

### 2.5. Data Analysis

Descriptive statistics, including percentages and 95% confidence intervals, were em-ployed to assess the epidemiological characteristics. Pearson’s Chi-square test was uti-lized to compare toxicity symptoms among patients who received either docetaxel or paclitaxel. All statistical analyses were carried out using SPSS software version 20, and a significance level of *p*-value ≤ 0.05 was deemed as indicating statistical significance.

## 3. Results

The majority of patients were in the 40–49 age range (32.7%), followed by 50–59 (26.3%) and 30–39 (14.7%). Brown ethnicity accounted for the highest percentage (66%), while white and black women represented 27.3% and 6.7%, respectively. Marital status showed a higher prevalence of married women (42.7%). In terms of education, most patients completed elementary school (43.7%). The largest occupational group consisted of women without employment (43.6%), followed by those engaged in autonomous work (15.7%) (Table 1).

The prevalence of smoking was 30.7%, while alcoholism was reported in 41.3% of cases. A sedentary lifestyle was observed in 76% of participants. The majority of women had their first pregnancy between the ages of 18–21 (36%), and 81.3% breastfed their children, with 82% having two or more children. Most women experienced menarche between the ages of 12 and 13 (47.7%). Menopause occurred predominantly between 50 and 60 years (25.3%), and 41.7% were pre-menopausal at the time of the study. A family history of cancer was reported by 64.7% of participants (Table 2).

Most cases (94.7%) exhibited non-special-type invasive ductal carcinoma. Among the molecular subtypes, Luminal B was the most prevalent (52%), followed by HER2+ (22.3%) and triple-negative (13%). In terms of disease progression, 76.3% of the patients had no reported metastasis or metastasis, while bone metastasis was observed in 8.3% of cases. Other sites of recurrence included the plastron (1.3%), contralateral breast (3.4%), lung (3.4%), and brain (0.6%). Metastasis involving multiple sites occurred in 6.7% of cases (Table 3).

Considering the treatment administered, among the female participants in this study, after undergoing taxane chemotherapy, 82.7% experienced at least one type of toxicity that they had never experienced before the start of this treatment. Among these, neurotoxicity was the most frequent (52.7%), followed by gastrointestinal toxicity (50%) and musculoskeletal toxicity (45%). Infusion reactions were observed in 33.7%, myelotoxicity in 12%, and only 11.3% reported having experienced dermatological toxicity (Table 4).

When specifying the taxane used, paclitaxel and docetaxel, in comparison to the two treatment groups, it was possible to highlight certain gastrointestinal symptoms such as diarrhea and mucositis, as well as infusion reactions (such as hypersensitivity reactions and severe back pain during taxane infusion). Diarrhea was significantly more prevalent in the Docetaxel group (36.6%) compared to the Paclitaxel group (19.3%). 

The incidence of mucositis was notably higher in the Docetaxel group (5.3%) in con-trast to the Paclitaxel group (0.7%). On the other hand, hematological toxicities, such as anemia and neutropenia, were significantly more prevalent in the Paclitaxel group. Addi-tionally, the occurrence of infusion reactions was significantly greater in the Docetaxel group (44%) compared to the Paclitaxel group (23.3%) (Table 5).

## 4. Discussion

The majority of patients in the Brazilian Amazon region undergoing chemotherapy treatment for breast cancer (BC) were in the 40–49 age range. This finding is consistent with a study conducted in Mexico, which included 4300 women with BC, and reported a mean age at diagnosis of 52 years, with 15.3% of women being under 40 years old [9]. Advanced age, particularly those over 50, is the most common stage of life for BC diagnosis, as cumulative exposures throughout life and biological changes associated with aging increase the risk of the disease [10].

Regarding marital status in our study, 42.7% of participants reported being married, which aligns with another similar study conducted in Brazil that found a majority were also married, albeit at a slightly higher rate of 62.9% [11]. When comparing ethnicity and marital status, a study conducted in California evaluated the impact of marital status and the role of race/ethnicity in mortality among women with triple-negative breast cancer (TNBC). This study concluded that marital status did not influence the risk of mortality for black women; however, separated and widowed white women had a higher risk of mortality than married white women [12].

The epidemiological profile related to occupation and education may be associated with the high incidence and mortality of breast cancer in low- or middle-income countries [9,10]. In our results, the majority of women (43.7%) reported having no occupation, which is supported by a study conducted in Central Africa [13], where 48.9% of women were housewives. This differs from another Brazilian study, where the majority (25.6%) were rural workers (farmers), and only 19.2% had no occupation [14].

Studies have shown that smoking can be associated with an increased risk of developing BC. However, the evidence regarding this association is complex and sometimes contradictory. The International Agency for Research on Cancer (IARC) classifies smoking as a carcinogenic agent with limited evidence for BC in humans [15]. Smoking is known to contain numerous harmful chemicals, including carcinogens, that can potentially damage DNA and increase the risk of cancer development. However, the mechanisms by which smoking influences BC risk are not fully understood. Our studies have found only 30.7% of patients smoke and it was similar to other research [8,9,10,11,12,13,14,15,16].

Studies have demonstrated a positive association between alcohol intake and the incidence of breast cancer. The mechanism behind this association is believed to be related to the effect of alcohol on hormone levels, particularly estrogen. Alcohol can increase estrogen levels in the body, which can promote the growth of hormone-sensitive breast cancer cells. Additionally, alcohol may interfere with the body’s ability to metabolize and eliminate carcinogens, further contributing to the development of breast cancer [10]. In our study, 41.3% of women with BC reported being an alcoholic or former alcoholic, corroborating other studies in the Brazilian Amazon region [8,9,10,11,12,13,14,15,16].

The majority of women (76%) reported a sedentary lifestyle without physical activity, which increases the risk of breast cancer. Studies show that physical activity reduces this risk through various mechanisms, such as reducing inflammation markers and estrogen levels, improving insulin resistance, and enhancing immune function. [17]. A UK study with over 170,000 women demonstrated a protective association between physical activity and breast cancer risk, with a 21% risk reduction in pre-menopausal and 16% in postmenopausal women, regardless of any association of risk from its effects on obesity [18].

Other risk factors have been identified for breast cancer (BC), including age at first pregnancy, age at menarche, and age at menopause. Long menstrual history (early menarche and late menopause), late primiparity (after 30 years), or nulliparity are associated with an increased risk of BC [19]. Contrasting these data, our study showed that most women with BC had their first pregnancy between the ages of 18–21, and experienced menarche between the ages of 12 and 13 and menopause occurred predominantly between 50 and 60 years. Other studies also obtained similar results to ours in Morocco [20] and in Mexico [9].

Breastfeeding is considered a protective factor for BC due to the hypoestrogenic state of women during the breastfeeding period [21]. A clinical and epidemiological profile study that included 4300 women with breast cancer treated in Mexico City found that most women (63%) reported not having breastfed their children [9]. In another epidemiological profile study of women with breast cancer in Brazil, they reported that more than 80% of BC patients breastfed their children [16]. In our results, the majority of women with BC (81.3%) reported having breastfed, and 66.7% breastfed two or more children.

Family history is a recognized significant risk factor for breast cancer (BC), with hereditary susceptibility attributed to mutations in genes like BRCA1 and BRCA2 [15]. In our study, 64.7% of the participants reported a family history of cancer, consistent with other research where BC was the predominant cancer in affected families [22]. A study in the United Kingdom involving over 113,000 women revealed that those with a first-degree relative with BC had a 1.75-fold higher risk of developing the disease compared to women without a family history. The risk increased to 2.5 times for women with two or more first-degree relatives with BC [23].

When examining the clinical profile of women diagnosed with breast cancer (BC), in line with findings from various global studies [9,13,20,24], our results indicated that the predominant histological subtype was invasive carcinoma of no special type, previously referred to as invasive ductal carcinoma (IDC), accounting for 94.7%. This observation aligns with existing literature, which consistently identifies this subtype as the most prev-alent among all invasive breast carcinomas [25].

In the immunohistochemical analysis, the most frequent molecular subtype was Luminal B (52%), similar to the study in Morocco [20], where the analysis revealed that 46% of participants belonged to this subtype. In our study, only 12.7% were Luminal A, which is considered to have a better prognosis and higher survival rates when compared to Luminal B. It is important to identify Luminal B in BC because this subgroup is associated with a poorer prognosis among the luminal types and may benefit from additional local and systemic treatment [26].

Understanding the molecular subtype of breast cancer patients is crucial because it dictates various therapeutic approaches and specific clinical outcomes. This leads to per-sonalized treatment for each patient based on their molecular characteristics. Currently, precision medicine for breast cancer encompasses diagnosis, treatment, and disease pre-vention, all while considering each patient’s genetic composition. In terms of treatment, precision medicine offers the prospect of individualization. This means that every diag-nosed breast cancer patient receives the most suitable diagnoses and targeted therapies tailored to their disease profile, significantly enhancing patient survival and reducing healthcare system costs [27].

One notable discovery from our research highlights that 23.7% of the patients in our study cohort presented with metastatic tumors, with a predominant occurrence of bone metastasis. This finding is consistent with a study conducted in Morocco, which docu-mented a 29.5% incidence of bone metastasis among breast cancer patients [20]. Moreover, our study’s outcomes harmonize with those of a separate investigation in Mexico, where a noteworthy 36% of the participants were diagnosed with metastatic disease [9].

Regarding metastases, in a clinical characterization study of metastatic breast cancer patients who survived for more than ten years, approximately 70% of patients initially presented with metastases in a single organ, with no significant variation among subtypes. The lung was the most common site of recurrence, affecting 46.4% of all patients and 61.1% of patients in the hormone receptor-positive/HER2 subgroups. This was followed by recurrence only in distant lymph nodes (37.3%), which was more common in the HER2+ subgroups, and bone metastasis in 30% of all study patients [28]. Unfortunately, a significant number of cancer cases was diagnosed at an advanced stage in developing countries. Thus, delays in cancer diagnosis and limited access to treatment were important challenges for effective cancer control [20].

Concerning treatment, the administration of antineoplastic agents can result in a range of toxicities with varying antitumor efficacies, emphasizing the importance of individualized therapy. Even with prophylactic measures, taxanes still lead to many limiting toxicities, with neurotoxicity being one of the most common [6]. In our study, the vast majority (82.7%) experienced taxane-related toxicities, with peripheral neuropathy being the most common, affecting 44.7% of patients receiving docetaxel and 56% among those receiving paclitaxel. In a study conducted in Iran, out of 346 breast cancer patients undergoing taxane chemotherapy, 23.7% developed peripheral neuropathy, and significant negative correlations were observed between overall health status/quality of life, physical functioning, and role performance with the degree of neuropathy experienced [29].

Chemotherapy-related peripheral neuropathy can restrict treatment for many pa-tients. In a retrospective cohort study that included women undergoing treatment with docetaxel or paclitaxel for breast cancer at an Oncology Center affiliated with the Univer-sity of Pennsylvania Health System, it was observed that chemotherapy dosages were limited due to peripheral neuropathy, significantly in the case of paclitaxel, especially in those who received a weekly treatment regimen. Women who had their dosage reduced or discontinued received less cumulative chemotherapy than planned [30].

When specifying the Taxane used, the toxicities of greatest significance were gastro-intestinal, such as diarrhea and mucositis, and hematological, including anemia, neutro-penia, and infusion reactions. In a study conducted in Tunisia to analyze Taxane toxici-ties in Tunisian patients and determine their impact on treatment response, grade I-II gas-trointestinal toxicity was observed in 54.4% of cases, with hematological toxicities being less frequent but affecting dose reduction and treatment delays for patients [31].

In the management of hematological toxicities, tests such as complete blood counts and liver function tests are generally ordered during taxane treatment every 3 weeks to detect changes such as neutropenia, thrombocytopenia, anemia, and elevated liver transaminases. There are no established guidelines for routine blood tests in weekly drug regimens, and institutional policies should be followed [32].

In our study, infusion reactions (such as hypersensitivity reactions) were significant-ly more frequent in women who received Docetaxel. Taxanes are a class of antineoplastic drugs that often cause hypersensitivity reactions in cancer patients. These reactions typi-cally occur at the beginning of chemotherapy infusion, within the first 10 min during the first or second infusion. Minor reactions are observed in 40% of cases and result in hives or skin rash. Significant reactions include signs and symptoms such as nausea, vomiting, flushing, hives, dyspnea, wheezing, tachycardia, hypotension, and abdominal or chest pain [33].

To better manage these toxicities, guiding patients and their families regarding po-tential adverse effects associated with Taxanes is essential. They should be informed about lifestyle changes, such as regular use of sunscreen and avoiding sun exposure to prevent paclitaxel-induced dermatitis. Patients should also be educated about engaging in physical exercise, which can help with neuropathy. Furthermore, they should receive guidance on recognizing the signs and symptoms of febrile neutropenia, so they can seek care at the oncology emergency service when necessary, among other recommendations. In this regard, the multidisciplinary team, including nurses, pharmacists, and physio-therapists, plays a crucial role in educating patients and their families [32].

Toxicities can lead to the discontinuation of Taxane treatments, compromising pa-tient benefits, and are more frequently caused by severe toxicities than by disease progres-sion itself. Currently, several strategies for preventing these toxicities are being investigat-ed, including studies on genetic variants associated with chemotherapy-related toxicities, aiming for better therapeutic planning [34].

## 5. Conclusions

In conclusion, this study provides valuable information about the clinical–epidemiological profile, risk factors, and treatment toxicities in women with breast cancer undergoing taxane therapy in the Brazilian Amazon region. The clinical–epidemiological profile consisted of women aged over 40, of mixed race, were married and unemployed, and had a basic education, alcohol consumption habits, and physical inactivity. Many of them had a family history of cancer. Regarding the reproductive profile, the age of menarche was 12–13 years, menopause occurred after the age of 50, the first childbirth typically took place between the ages of 18 and 21, and breastfeeding multiple children was common. These women were diagnosed with invasive carcinoma of no special type and the Luminal B subtype, often experiencing multiple toxicities during taxane chemotherapy, with a particular emphasis on neurotoxicity. These findings underscore the importance of early detection, comprehensive assessment of risk factors, and effective management of treatment-related toxicities to improve the outcomes of breast cancer patients in this region.

The data obtained can contribute to the establishment of public policies, emphasizing the importance of implementing health education initiatives for early diagnosis and promoting healthy lifestyle habits, aiming to reduce morbidity and mortality while improving disease prognosis. Regarding toxicities, it underscores the importance of conducting more specific studies on genetic variants and identifying biomarkers that can enable early intervention, leading to individualized treatment, optimizing patient clinical outcomes, and enhancing quality of life, particularly in mixed populations, such as the Amazonian population.

## Figures and Tables

**Table 1 jpm-13-01458-t001:** Epidemiological profile of women with breast cancer receiving chemotherapy with taxanes in the Amazon region.

Characteristics	N (%)	CI 95%
Age range (years)		
24 to 29	6 (2.0%)	0.7–3.7
30 to 39	44 (14.7%)	10.7–18.7
40 to 49	98 (32.7%)	27.3–38.0
50 to 59	79 (26.3%)	21.3–31.7
60 to 69	53 (17.7%)	13.3–22.0
70 to 83	20 (6.6%)	4.0–9.7
Ethnicity *		
White	82 (27.3%)	22.0–32.3
Black	20 (6.7%)	3.7–9.3
Brown	198 (66%)	61.0–71.3
Marital status		
Single	88 (29.3%)	23.7–34.7
Married	128 (42.7%)	37.0–48.0
Stable union	46 (15.3%)	11.3–19.7
Divorced	17 (5.7%)	3.0–8.3
Widow	21 (7.0%)	4.3–10.0
Education		
lliterate	8 (2.7%)	1.0–4.7
Elementary School	131 (43.7%)	38.0–49.3
High school	127 (42.3%)	36.7–48.0
University education	34 (11.3%)	8.0–15.3
Profession/Occupation		
Rural activity	29 (9.7%)	6.7–13.0
Retired	20 (6.7%)	4.0–9.7
Autonomous	47 (15.7%)	11.7–20.0
Housekeeper	17 (5.7%)	3.3–8.3
Education Professional	17 (5.7%)	3.3–8.3
Health professional	15 (5%)	2.7–7.7
Without occupation	131 (43.6%)	38.3–49.7
Others	24 (8%)	5.0–8.0

* Self—declared ethnicity.

**Table 2 jpm-13-01458-t002:** Breast cancer risk factors in women in the Amazon region.

Characteristics	N = 300	CI 95%
Life Habits		
Smoking	92 (30.7%)	25.3–36.3
Alcoholism	124 (41.3%)	36.3–46.7
Sedentary lifestyle	228 (76%)	19.–29.0
Age at first pregnancy (years)		
13 to 17	52 (17.3%)	13.3–21.3
18 to 21	108 (36%)	30.3–42.0
22 to 25	49 (16.3%)	12.7–21.0
26 to 29	34 (11.3%)	8.0–15.0
30 to 42	31 (10.4%)	6.7–13.7
nulliparous	26 (8.7%)	5.7–12.0
Breast—feeding		
No	56 (18.7%)	14.7–23.0
Yes	244 (81.3%)	77.0–85.3
1 son 2 or more children	44 (18.0%) 200 (82.0%)	13.5–23.0 77.0–86.5
Age at Menarche (years)		
8 to 11 years	42 (14%)	10.3–18.3
12 to 13	143 (47.7%)	42.0–53.0
14 to 15	90 (30%)	25.0–35.3
>15	25 (8.3%)	5.3–11.7
Age at Menopause (years)		
40 to 49 years old	67 (22.3%)	18.0–27.0
50 to 60 years	76 (25.3%)	20.7–30.0
pre—menopausal	125 (41.7%)	36.0–47.3
Hysterectomy before cancer	32 (10.7%)	7.3–14.3
Family history of cancer		
No	106 (35.3%)	30.0–41.0
Yes	194 (64.7%)	59.0–70.0

**Table 3 jpm-13-01458-t003:** Clinical profile of women with breast cancer in the Amazon region.

Characteristics	N = 300	CI 95%
Histological Type		
Non—special type invasive breast carcinoma	284 (94.7%)	92.0–96.7
Special type carcinoma–invasive lobular carcinoma	11 (3.8%)	1.7–6.0
Other Special Type Carcinomas	5 (1.5%)	0.3–3.0
Molecular Subtype		
Luminal A	38 (12.7%)	8.7–16.7
Luminal B	156 (52%)	46.7–57.7
HER2+	67 (22.3%)	17.7–27.0
Triple negative	39 (13%)	9.3–17.0
Disease progression		
No metastasis or metastasis not reported	229 (76.3%)	71.7–28.3
Recurrence in the plastron	4 (1.3%)	0.3–2.7
Recurrence in the contralateral breast	10 (3.4%)	1.3–5.3
Bone metastasis	25 (8.3%)	5.3–11.7
Lung metastasis	10 (3.4%)	1.3–5.3
Brain metastasis	2 (0.6%)	0.0–1.7
Metastasis from more than one site (bone, liver, lung, brain)	20 (6.7%)	4.0–9.7

**Table 4 jpm-13-01458-t004:** Taxane toxicities in women with breast cancer in the Amazon region.

Toxicities	N = 300	CI 95%
No	52 (17.3%)	13.0–21.7
Yes	248 (82.7%)	78.3–87.0
One	48 (19.4%)	11.7–20.7
Two	84 (33.9%)	23.0–33.0
Three	77 (31%)	20.3–30.7
four or more	39 (15.7%)	9.3–17.0
Type of Toxicity *		
Neurological	158 (52.7%)	46.7–58.3
Gastrointestinal	150 (50%)	44.3–55.7
Musculoskeletal	135 (45%)	39.3–50.3
Infusion Reaction	101 (33.7%)	28.3–39.0
Hematological	36 (12.0%)	8.3–15.7
Dermatological	34 (11.3%)	7.7–15.0

* except for alopecia, which is related to other antineoplastic treatments previously performed by the patients, such as Doxorubicin.

**Table 5 jpm-13-01458-t005:** Docetaxel and Paclitaxel toxicity in women with breast cancer in the Amazon region.

Toxicity	Docetaxel (N = 150)	Paclitaxel (N = 150)	*p*-value ^a^
Gastrointestinal			
Diarrhea	55 (36.6%)	29 (19.3%)	0.0013 *
Constipation	5 (3.3%)	11 (7.3%)	0.1989
Nausea and vomiting	44 (28.3%)	34 (22.7%)	0.2362
Anorexia/Lack of appetite	11 (7.3%)	9 (6.0%)	0.8170
Mucositis	8 (5.3%)	1 (0.7%)	0.0195 *
Neurological			
Peripheral neuropathy	67 (44.7%)	84 (56.0%)	0.0647
Headache	7 (4.7%)	3 (2.0%)	0.4346
Insomnia	1 (0.7%)	7 (4.7%)	0.0732
Musculoskeletal			
Arthralgia	51 (34%)	44 (29.3%)	0.4365
Myalgia	49 (32.7%)	34 (22.7%)	0.0708
Ostealgia	22 (14.7%)	24 (16%)	0.8727
Dermatological			
Skin disorders	7 (4.7%)	7 (4.7%)	1.000
Nail disorders	13 (8.7%)	6 (4.0%)	0.1549
Body itch	6 (4.0%)	2 (1.3%)	0.2823
Hematological			
Anemia	6 (4.0%)	20 (13.3%)	0.0076 *
Neutropenia	6 (4.0%)	19 (12.7%)	0.0122 *
Thrombocytopenia	3 (2.0%)	7 (4.7)	0.3346
Infusion Reaction Yes No	66 (44%) 84 (56%)	35 (23.3%) 115 (76.7%)	0.0002 *

^a^ Chi-square Test. *. *p*-value < 0.05.

## Data Availability

The data presented in this study are openly available on Figshare at https://doi.org/10.6084/m9.figshare.23929074.v1; accessed on 10 August 2023.

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
