# Peer review of "Breast Cancer: Clinical–Epidemiological Profile and Toxicities of Women Receiving Treatment with Taxanes in the Amazon Region"

_jpm, 2023, doi:10.3390/jpm13101458_

Round 1
Reviewer 1 Report
Rewrite the Abstract as per the suggestions in the manuscript.
In materials and methods:
1. Describe in details about the patients combinations chemotherapy with Texanes in detail.
2. Summary of Treatment profile of the patients.
3. Laos describe the analysis parameters.
Results:
1. details of the other chemotherapy or any other therapy need to be addressed as this will avoid the misleading of the conclusion about taxanes alone.
2. Table 4 and Table 5 : please elaborate how these conditions due to taxanes not due to any other reasons?
Discussion:
Need to be more elaborated and should be discussed with earlier findings.
Need to be improved and there are few typo which need to be addressed.
Author Response
|
Response to Reviewer X Comments
|
||
|
1. Summary |
|
|
|
Thank you very much for taking the time to review this manuscript. Please find the detailed responses below and the corresponding revisions/corrections highlighted/in track changes in the re-submitted files.
|
||
|
2. Questions for General Evaluation |
Reviewer’s Evaluation |
Response and Revisions |
|
Does the introduction provide sufficient background and include all relevant references? |
Can be improved |
The introduction was rewritten with relevant information.
|
|
Are all the cited references relevant to the research? |
Can be improved |
The references were improved and relevant to the research.
|
|
Is the research design appropriate? |
Must be improved |
The research design was improved.
|
|
Are the methods adequately described? |
Must be improved |
Methods were rewrited conseidering the suggestions of the reviewer.
|
|
Are the results clearly presented? |
Yes |
The results were improved.
|
|
Are the conclusions supported by the results? |
Must be improved |
The conclusion was improved according to the results.
|
|
3. Point-by-point response to Comments and Suggestions for Authors |
||
|
Comments 1: Rewrite the Abstract as per the suggestions in the manuscript.
|
||
|
Response 1: Thank you for pointing this out. The abstract was rewritten according to the suggestions of the manuscript in page number 01 and line 15 to 30. |
||
|
Comments 2: In materials and methods: 1. Describe in details about the patients combinations chemotherapy with Texanes in detail. 2. Summary of Treatment profile of the patients. 3. Laos describe the analysis parameters.
|
||
|
Response 2: We Agree. The materials and methods were rewritten as suggested by the reviewer. The combinations of chemotherapy were described on page 2, in lines 79 to 81.The summary of the treatment profile was described on page 2, in lines 81 to 87. The parameters for toxicity analysis were described on pages 2 and 3, in lines 93 to 102.
Comments 3: Results: 1. details of the other chemotherapy or any other therapy need to be addressed as this will avoid the misleading of the conclusion about taxanes alone. 2. Table 4 and Table 5 : please elaborate how these conditions due to taxanes not due to any other reasons? Response 3: The results were described as suggested by the reviewer. The details of the therapies were described in the material and method, on pages 2 and 3, in lines 93 to 102. The signs and symptoms of toxicities were not reported by the women in the study before the start of treatment with Taxanes, which was described on page 6, in the 134 to 141.
Comments 4: Discussion: Need to be more elaborated and should be discussed with earlier findings. Response 4: The discussion has been expanded as suggested, page 10, in lines 252 to 276.
|
||
|
4. Response to Comments on the Quality of English Language |
||
|
Point 1: Need to be improved and there are few typo which need to be addressed. |
||
|
Response 1: The manuscript was reviewed by an American Native Speaker.
|
||
|
5. Additional clarifications |
||
|
We rewrote the manuscript with 4000 + word count as suggested. Also, the number of references cited in the text was greater than 30. |
||
Reviewer 2 Report
Author Marta Solange Camarinha Ramos Costa and colleagues have submitted an article focusing on the clinical-epidemiological profile and toxicities among breast cancer patients receiving taxanes as treatment in the Amazon region. While the adverse effects of docetaxel and paclitaxel as long-term treatment modalities have been discussed in studies before, the present study presents the outcomes from an epidemiological perspective which makes it a significant study. The statistical tests and data interpretation were correctly performed.
Author Response
|
Response to Reviewer X Comments
|
||
|
1. Summary |
|
|
|
Thank you very much for taking the time to review this manuscript. Please find the detailed responses below and the corresponding revisions/corrections highlighted/in track changes in the re-submitted files.
|
||
|
2. Questions for General Evaluation |
Reviewer’s Evaluation |
Response and Revisions |
|
Does the introduction provide sufficient background and include all relevant references? |
Yes |
|
|
Are all the cited references relevant to the research? |
Yes |
|
|
Is the research design appropriate? |
Yes |
|
|
Are the methods adequately described? |
Yes |
|
|
Are the results clearly presented? |
Yes |
|
|
Are the conclusions supported by the results? |
Can be improved |
The conclusion was improved according to the results.
|
|
|
||
|
|
||
|
3. Point-by-point response to Comments and Suggestions for Authors Comments 1: Author Marta Solange Camarinha Ramos Costa and colleagues have submitted an article focusing on the clinical-epidemiological profile and toxicities among breast cancer patients receiving taxanes as treatment in the Amazon region. While the adverse effects of docetaxel and paclitaxel as long-term treatment modalities have been discussed in studies before, the present study presents the outcomes from an epidemiological perspective which makes it a significant study. The statistical tests and data interpretation were correctly performed. Response 1: We Thank you for your comments.
4. Additional clarifications |
||
|
We rewrote the manuscript with 4000 + word count as suggested. Also, the number of references cited in the text was greater than 30. |
||